# A regulatory pathway that selectively up-regulates elongasome function in the absence of class A PBPs

**Yesha Patel, Heng Zhao†, John D Helmann\***

Department of Microbiology, Cornell University, Ithaca, United States

**Abstract** Bacteria surround themselves with peptidoglycan, an adaptable enclosure that contributes to cell shape and stability. Peptidoglycan assembly relies on penicillin-binding proteins (PBPs) acting in concert with SEDS-family transglycosylases RodA and FtsW, which support cell elongation and division respectively. In *Bacillus subtilis*, cells lacking all four PBPs with transglycosylase activity (aPBPs) are viable. Here, we show that the alternative sigma factor $\sigma^I$ is essential in the absence of aPBPs. Defects in aPBP-dependent wall synthesis are compensated by $\sigma^I$-dependent upregulation of an MreB homolog, MreBH, which localizes the LytE autolysin to the RodA-containing elongasome complex. Suppressor analysis reveals that cells unable to activate this $\sigma^I$ stress response acquire gain-of-function mutations in the essential histidine kinase WalK, which also elevates expression of *sigI*, *mreBH* and *lytE*. These results reveal compensatory mechanisms that balance the directional peptidoglycan synthesis arising from the elongasome complex with the more diffusive action of aPBPs.

**\*For correspondence:**
jdh9@cornell.edu

**Present address:** †Department of Microbial Pathogenesis, Yale University School of Medicine, New Haven, United States

**Competing interests:** The authors declare that no competing interests exist.

## Introduction

Nearly all bacterial cells are surrounded by a peptidoglycan (PG) cell wall that provides a protective barrier, helps resist cell swelling and lysis under hypoosmotic conditions, and contributes to cell shape determination (*Egan et al., 2020*; *Zhao et al., 2017*). PG functions as a large, covalently linked macromolecular enclosure and is actively remodeled to allow cell growth and division. The basic processes of PG synthesis are broadly conserved, and the detailed pathways are well documented. PG synthesis initiates with the diversion of sugars from central metabolism to form the two amino-sugars, N-acetylglucosamine (NAG) and N-acetylmuramic acid (NAM), and the incorporation of amino acids to form the stem peptide (*Barreteau et al., 2008*). The ultimate product of these cytosolic reactions is lipid II, a disaccharide pentapeptide precursor unit linked to an undecaprenyl pyrophosphate carrier lipid (*van Heijenoort, 2007*). Lipid II is flipped across the membrane (*Sham et al., 2014*; *Meeske et al., 2015*) where it interacts with two key enzymatic activities to assemble the PG layer: a transglycosylase (TG) function joins the disaccharide unit to form long, linear chains of alternating NAG-NAM residues, and a transpeptidase (TP) activity crosslinks a subset of the pentapeptide side chains to link the glycan strands together. Crucially, insertion of new glycan strands requires endopeptidases that can cleave existing crosslinks to facilitate cell wall expansion (*Singh et al., 2012*; *Hashimoto et al., 2012*; *Do et al., 2020*).

Most bacteria require PG for survival, except under very specific conditions (*Claessen and Errington, 2019*). This, combined with the absence of PG in eukaryotes, makes PG synthesis and stability an excellent target for antibiotics. One class of PG-targeting antibiotics, the beta-lactams, account for more than 60% of the global market (*Klein et al., 2018*). Beta-lactam antibiotics interfere with PG synthesis by covalently modifying penicillin-binding proteins (PBPs), named for their affinity for the first widely used member of this drug family. All PBPs have TP activity, and beta-lactams mimic the substrate of the transpeptidation reaction (*Tipper and Strominger, 1965*). Many PBPs also have

TG activity, and these bifunctional PBPs are designated class A PBPs, or aPBPs (*McPherson and Popham, 2003*). Other PBPs, designated bPBPs, only have TP activity, and must work in coordination with enzymes that provide TG activity (*Wei et al., 2003*; *Taguchi et al., 2019*; *Rohs et al., 2018*; *Özbaykal et al., 2020*).

While the basic outline of PG assembly has been understood for decades, the last few years have seen major strides in our understanding of how PG synthesis is coordinated in time and space (*Zhao et al., 2017*; *Egan et al., 2020*). Moreover, PG synthesis can be regulated as a function of cell growth, division, nutritional status, and in response to externally imposed stresses such as the action of antibiotics (*Delhaye et al., 2019*; *Typas et al., 2012*; *Helmann, 2016*). *B. subtilis* has been a leading model system for understanding PG synthesis in rod-shaped, Gram-positive bacteria. Seminal work in this system established, for example, that the sites of PG synthesis during cell elongation seem to be correlated with cytoskeletal filaments assembled from MreB and its paralogs, MreBH and Mbl (*Kawai et al., 2009*). This synthesis occurs in arcs that are perpendicular to the long access of the cell and is driven by a putative complex known as the elongasome (*Garner et al., 2011*). Cell division, in contrast, occurs at mid-cell during vegetative growth and is directed by a different cytoskeletal filament, FtsZ, in a complex called the divisome (*Mahone and Goley, 2020*). In early models, it was suggested that the major aPBP, PBP1 (encoded by the *ponA* gene), shuttled between the elongasome and divisome to provide the needed TG and TP activities (*Claessen et al., 2008*). However, bPBPs clearly also play important roles in synthesis (*Wei et al., 2003*). The composition and dynamic nature of these complementary systems has been subject of intensive study.

A key finding that challenged our understanding of PG synthesis in *B. subtilis* was the observation that a strain lacking all four known aPBPs was viable and still synthesized an apparently normal PG layer (*McPherson and Popham, 2003*). This implied that there must be another protein with TG activity and, unlike aPBP-associated TG activity, this activity was insensitive to inhibition by moenomycin (MOE). MOE, like many PG synthesis inhibitors, activates the $\sigma^M$ stress response (*Mascher et al., 2007*). Moreover, *sigM* null mutants are highly MOE sensitive (*Mascher et al., 2007*), which suggested that the missing TG might be part of the $\sigma^M$ regulon. Indeed, the elongasome-associated TG has been identified as the SEDS family protein RodA (*Meeske et al., 2016*; *Emami et al., 2017*), a known member of the $\sigma^M$ regulon (*Eiamphungporn and Helmann, 2008*; *Helmann, 2016*). A RodA paralog, FtsW, provides TG activity in the context of the divisome (*Taguchi et al., 2019*; *Liu et al., 2018*).

Our current understanding of PG synthesis during cell elongation in *B. subtilis* suggests that the bulk of synthesis is provided by the elongasome, with RodA serving as TG and PBP2a and PbpH, and perhaps also aPBPs, serving as TP (*Emami et al., 2017*; *Meeske et al., 2016*). This action is directional, largely oriented perpendicular to the long cell axis, and is balanced by a more diffusive activity of aPBPs (*Dion et al., 2019*; *Vigouroux et al., 2020*). Cells that rely exclusively on the elongasome for growth are longer and thinner, whereas those that rely predominantly on aPBPs tend to be wider and shorter (*Dion et al., 2019*). Many PG synthesis inhibitors activate the $\sigma^M$ regulon, and this leads to elevated expression of many key PG biosynthetic enzymes (MurB, Amj, BcrC), elongasome components (MreB, RodA, MreCD), and the major aPBP (PBP1) (*Eiamphungporn and Helmann, 2008*; *Helmann, 2016*). However, some antibiotics may act selectively on the aPBPs or the elongasome, and it is less clear how cells might act to balance these two biosynthetic activities.

Here, we sought to define pathways important for fitness in cells that rely exclusively on the elongasome for cell elongation. We demonstrate that cells lacking aPBPs, or even just PBP1 (*ponA*), require a regulatory pathway that selectively increases expression of elongasome-associated proteins. Specifically, Δ*ponA* mutant cells are unable to grow in the absence of $\sigma^I$, which induces transcription of genes encoding MreBH and an associated autolysin, LytE. Factors that facilitate $\sigma^I$ activity, including the RasP intramembrane peptidase and its regulator EcsAB, are therefore also essential under these conditions. Further support for the importance of MreBH and LytE derives from analysis of a suppressor mutation that activates the WalKR two-component system, and thereby also restores viability to a Δ*rasP*Δ*ponA* double mutant by up-regulating these same elongasome components. These results suggest that the $\sigma^I$ stress response acting in concert with the WalKR system helps to maintain balanced activity of the elongasome and the aPBPs during cell elongation.

## Results

### The EcsAB-RasP pathway is essential in the absence of class A PBPs

Bacteria often use overlapping or redundant systems to sustain essential pathways such as PG synthesis. To identify genes with significant roles in elongasome activity in *B. subtilis,* we constructed a strain (designated ∆4) lacking all four class A PBPs (aPBPs), and which therefore relies solely on the elongasome for PG synthesis during cell elongation (*McPherson and Popham, 2003*). A Tn-Seq approach was employed to identify genes essential in the ∆4 strain but not in the wild-type (WT) background. We identified the *ecsAB* operon as having numerous mariner transposon insertions in WT, but very few in the ∆4 strain (*Figure 1—figure supplement 1*). We verified conditional essentiality of *ecsA* by determining the plating efficiency of a clean, unmarked deletion mutant (∆*ecsA*) in a *ponA* depletion background in the presence and absence of the genes encoding the other 3 aPBPs (*pbpD*, *pbpF*, *pbpG*). Interestingly, *ecsA* was not only essential in the ∆4 background but also with depletion of *ponA* alone (*Figure 1A*). Mutations that impair PG synthesis can often be rescued by growth on plates amended with 20 mM $MgSO_4$, which leads to decreased activity of autolysins and thereby helps restore balance between PG synthesis and degradation pathways (*Formstone and Errington, 2005*). Indeed, an ∆*ecsA*∆*ponA* mutant was viable when streaked on high Mg plates, and growth was Mg-dependent (*Figure 1B*).

EcsA has been designated as part of an ABC-type transporter involved in the expression and secretion of proteins (*Leskelä et al., 1999*). Deletion of *ecsA* has a profound effect on the intramembrane protease RasP, with similar phenotypes noted for the *ecsA* and *rasP* deletion mutants (*Heinrich et al., 2008*). Consequently, we tested whether the essential role of EcsA in the ∆*ponA* strain was due to RasP. Indeed, viability of ∆*rasP*∆*ponA*, like ∆*ecsA*∆*ponA*, depended on high Mg concentrations (*Figure 1B*). The above data highlight the importance of the EcsAB-RasP pathway in maintaining viability in the absence of aPBPs.

### Mutants defective in the EcsAB-RasP pathway are sensitive to antibiotics that inhibit aPBPs

Upregulation of elongasome activity is known to alleviate aPBP defects (*Meeske et al., 2016*). Based on the observed conditional essentiality, we hypothesized that the EcsAB-RasP pathway might functionally compensate for the absence of aPBPs. As a first test of this hypothesis, we measured sensitivity to moenomycin (MOE), a specific inhibitor of aPBP-associated TG activity (*Van Heijenoort et al., 1978*; *Chen et al., 2019*). Indeed, *ecsA* and *rasP* mutants were MOE sensitive with a four-fold decrease in minimum inhibitory concentration (MIC) relative to WT (*Table 1*). This was not due to a general growth defect: *ecsA* and *rasP* single mutants grew as well as WT in the absence of MOE, albeit with some lysis in stationary phase (*Figure 2A*), consistent with previous observations (*Heinrich et al., 2008*). This antibiotic sensitivity could be complemented by ectopic expression of *ecsAB* or *rasP*, respectively (*Figure 2—figure supplement 1*). Moreover, ∆*ecsA*∆*rasP* had a similar MOE sensitivity as ∆*rasP* (*Figure 2A*), suggesting that the synthetic lethality of *ecsA* with *ponA* is mediated through its known downstream effect on the activity of RasP (*Heinrich et al., 2008*). In contrast to MOE, the ∆*rasP* and ∆*ponA* mutants had a similar sensitivity as WT when tested for sensitivity to antibiotics that act on substrates common to both the elongasome and aPBP-dependent pathways of PG synthesis. For example, both nisin (*Wiedemann et al., 2001*) and vancomycin (*Watanakunakorn, 1984*) bind the common lipid II intermediate (*Figure 2—figure supplement 1*). Together, these results suggest that the EcsAB-RasP pathway is critical when aPBPs are compromised, but not as a general response to inhibition of PG synthesis.

We next sought to test antibiotics that, unlike MOE, inhibit aPBPs at their TP active site. We reasoned that a stress response important for elongasome activity should also provide resistance to antibiotics that inhibit aPBPs, assuming they do not also interfere with the bPBPs essential for the elongasome. We tested 4 β-lactams (cefuroxime, oxacillin, ampicillin and penicillin G) for their inhibition profiles against ∆*rasP* and ∆*ponA* strains. Oxacillin and cefuroxime (CEF) were previously suggested to preferentially inhibit aPBPs (*Sassine et al., 2017*; *Sharifzadeh et al., 2020*), whereas penicillin G preferentially inhibits bPBPs (*Sassine et al., 2017*). Consistently, oxacillin and CEF had highest activity against ∆*rasP*, whereas penicillin G and ampicillin had the highest activity against ∆*ponA*, which encodes the major aPBP, PBP1 (*Figure 2B*). These results support the idea that the

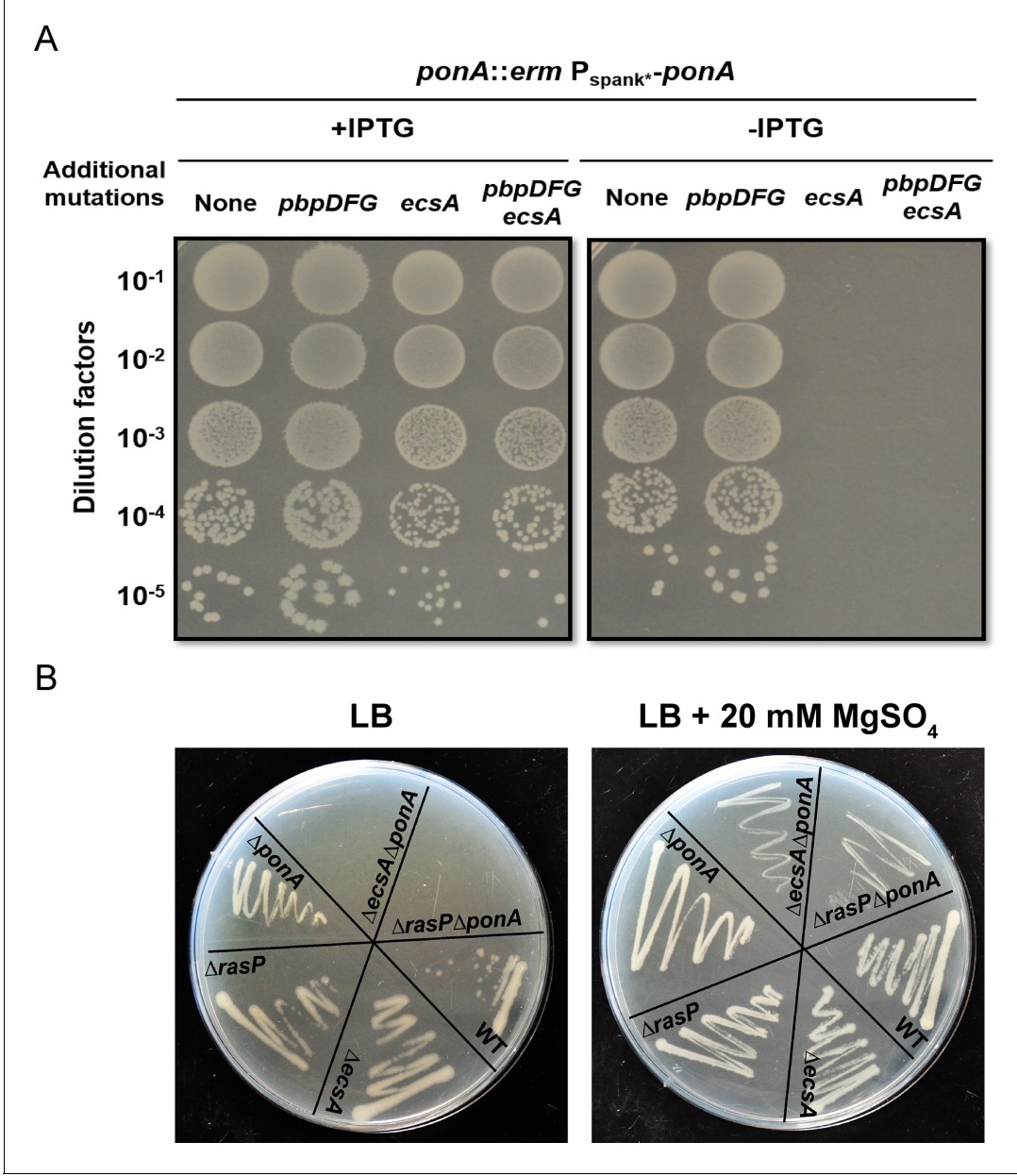

**Figure 1.** The *ecsA* and *ponA* genes are synthetic lethal in LB medium. (A) Plating efficiency of *ecsA* deletion mutants. Right panel: spot dilutions were used to assess the effect of an *ecsA* null mutation on growth in a *ponA* depletion background (-IPTG) with and without additional mutations in *pbpD*, *pbpF*, *pbpG* (to mimic the Δ4 A PBP background). Left panel: *ponA* was induced (+IPTG) from the $P_{spank*}$ promoter. (B) Growth of Δ*ecsA*, Δ*rasP*, Δ*ponA* and the double mutants Δ*ecsA*Δ*ponA* and Δ*rasP*Δ*ponA* on LB agar plates with and without supplementation with 20 mM MgSO₄.

The online version of this article includes the following figure supplement(s) for figure 1:

**Figure supplement 1.** Transposon insertion profile of the *ecsAB* operon.

EcsAB-RasP pathway functionally compensates either for the absence of aPBPs or for their chemical inhibition at either the TG (MOE) or TP (CEF) active sites.

Interestingly, the Δ*ponA* mutant was actually more CEF resistant than WT. Thus, PBP1 inactivated by CEF may be deleterious to the cell. This is suggestive of futile cycling, a process in which inactivation of the TP active site leads to an ongoing generation and degradation of uncrosslinked PG strands driven by the aPBP-associated TG (*Cho et al., 2014*; *Waxman et al., 1980*). To explore this idea further, we treated WT cells with sub-inhibitory concentrations of two drugs simultaneously, MOE and CEF, that inhibit the two different active sites of the aPBP proteins. If CEF results in futile

**Table 1.** Minimum inhibitory concentration (MIC) of various strains for moenomycin in μg/mL.

| Strains | Moenomycin MIC (μg/mL) |
|---|---|
| *WT* | 1.6 |
| Δ*ecsA* | 0.4 |
| Δ*rasP* | 0.4 |
| Δ*ponA* | >1.6 |
| Δ*sigW* | 1.6 |
| Δ*sigV* | 1.6 |
| Δ*sigI* | 0.4 |
| Δ*25ftsL* | 1.6 |

cycling, we reasoned that MOE might antagonize this effect. In contrast, MOE and CEF together resulted in synergistic inhibition (*Figure 2—figure supplement 2*). This is consistent with the *same target drug synergy model*, as previously described for *E. coli* protein synthesis inhibitors (*Yilancioglu, 2019*) and drugs used to treat human diseases (*Jia et al., 2009*), but does not support the hypothesis of CEF-dependent futile cycling.

## EcsAB-RasP functions through σ^I to sustain cell wall synthesis in the absence of aPBPs

RasP functions as an intramembrane protease for the activation of multiple stress response pathways, and our results suggest it may be important for PG synthesis when aPBPs are missing or inhibited. RasP proteolytically inactivates the anti-sigma factors RsiW (regulator of σ^W) (*Schöbel et al., 2004*), RsiV (regulator of σ^V) (*Hastie et al., 2013*) and RsgI (regulator of σ^I) (*Liu et al., 2017*). In the absence of RasP, these σ factors can not be activated. RasP also cleaves FtsL, a cell division protein (*Bramkamp et al., 2006*). To determine which of these RasP targets may contribute to elongasome activity, we took advantage of the fact that MOE and CEF selectively inactivate aPBPs. Therefore, MOE and CEF resistance provides a readout of elongasome function. We tested mutants lacking each of the three RasP-dependent sigma factors or containing Δ25FtsL, coding for a functional, but truncated FtsL (deleted in amino acids 2–26) variant that is not subject to cleavage by RasP (*Bramkamp et al., 2006*). The Δ*ecsA* and Δ*rasP* mutants were 4-fold more sensitive to MOE than

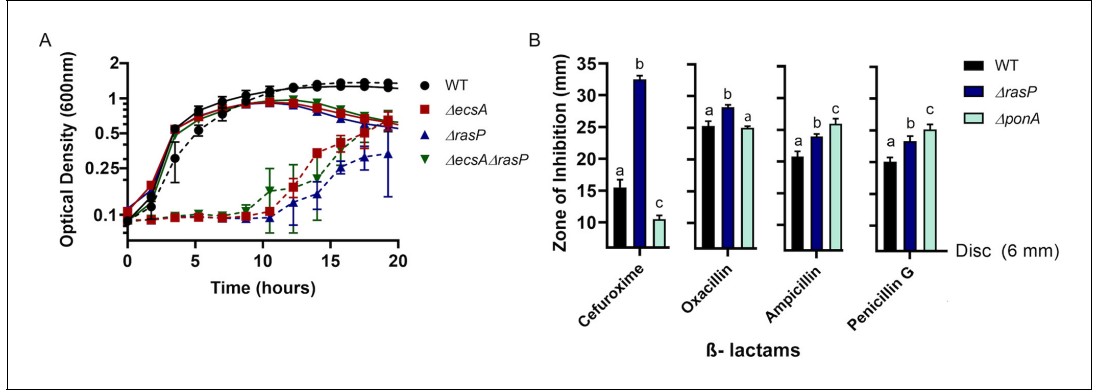

**Figure 2.** The EcsAB-RasP pathway is important for intrinsic antibiotic resistance. (**A**) Growth kinetics of WT, Δ*ecsA*, Δ*rasP* and the Δ*ecsA*Δ*rasP* double mutant in liquid LB medium with (dotted lines) and without (continuous lines) 0.4 μg/mL moenomycin (MOE). (**B**) β-lactam sensitivity of Δ*rasP* and Δ*ponA* strains determined by disc diffusion assay using cefuroxime (CEF) (10 μg), oxacillin (3 μg), ampicillin (15 μg), and penicillin G (20 units). No comparison was done between antibiotic groups. P-value cutoff of <0.001 was used.

The online version of this article includes the following source data and figure supplement(s) for figure 2:

**Source data 1.** Data of growth kinetics and zone of inhibition.
**Figure supplement 1.** Antibiotic susceptibility of Δ*ecsA* and Δ*rasP* mutants.
**Figure supplement 2.** Synergistic interaction of MOE and CEF in *B. subtilis*.

WT (0.4 vs. 1.6 µg/mL), whereas for Δ*ponA* the (MIC) was >1.6 µg/mL (*Table 1*; *Figure 3—figure supplement 1*). The MIC was unaffected by deletion of *sigW* or *sigV* or by the non-cleavable FtsL (1.6 µg/mL). However, the Δ*sigI* mutant was significantly more sensitive to MOE with the MIC being 0.4 µg/mL, similar to Δ*rasP*. This suggests that σ^I is required for optimal function of the MOE-insensitive elongasome.

Similar results were observed when CEF sensitivity was monitored (*Figure 3A*). Of the known RasP targets, σ^I contributes the most to CEF resistance. Moreover, the Δ*sigW*Δ*sigI* mutant phenocopies the Δ*rasP* mutant, suggesting that activation of σ^I and σ^W largely accounts for the role of RasP in CEF resistance. In addition, the sensitivity of the Δ*ecsA* and Δ*rasP* mutants was not further increased by mutation of *sigW* or *sigI* (*Figure 3—figure supplement 2*), indicative of them being in the same pathway. Finally, deletion of *rsgI*, encoding the σ^I anti-sigma factor, led to a significant decrease in CEF sensitivity of the Δ*ecsA* and Δ*rasP* mutants. Δ*rsgI* was more sensitive to CEF compared to WT, which may be due to increased activity of σ^I and its associated autolysins. In contrast, deletion of *rsiW*, encoding the σ^W anti-sigma factor, led to a much less pronounced effect (*Figure 3—figure supplement 2*). Thus, σ^I plays a dominant role in intrinsic CEF resistance, and as expected this activity relies on the RasP-dependent degradation of the RsgI anti-sigma factor.

The importance of σ^I in the absence of aPBPs was confirmed by determining the plating efficiency of Δ*sigI*Δ*ponA* double mutant (*Figure 3B*). The double mutant could survive with high Mg^{2+}, but was unable to grow on LB. This synthetic lethality of the Δ*sigI*Δ*ponA* and Δ*rasP*Δ*ponA* strains was suppressed by ectopically expressing the *sigI* gene from the leaky promoter P_{spac(hy)}. Thus, decreased σ^I activity can fully explain the Δ*rasP* antibiotic sensitivity phenotypes, and we therefore conclude that one or more members of the σ^I regulon must facilitate growth under conditions of impaired aPBP activity.

## σ^I supports elongasome function by regulating MreBH and LytE

Next, we sought to identify the σ^I-dependent genes important for survival in the absence of aPBPs. Of the genes directly regulated by σ^I (*Ramaniuk et al., 2018*), five (*mreBH*, *lytE*, *gsiB*, *fabI* and *bcrC*) have known or likely roles related to cell envelope functions. GsiB is a general stress response protein (*Michna et al., 2016*) and FabI is involved in fatty acid synthesis (*Heath et al., 2000*). BcrC

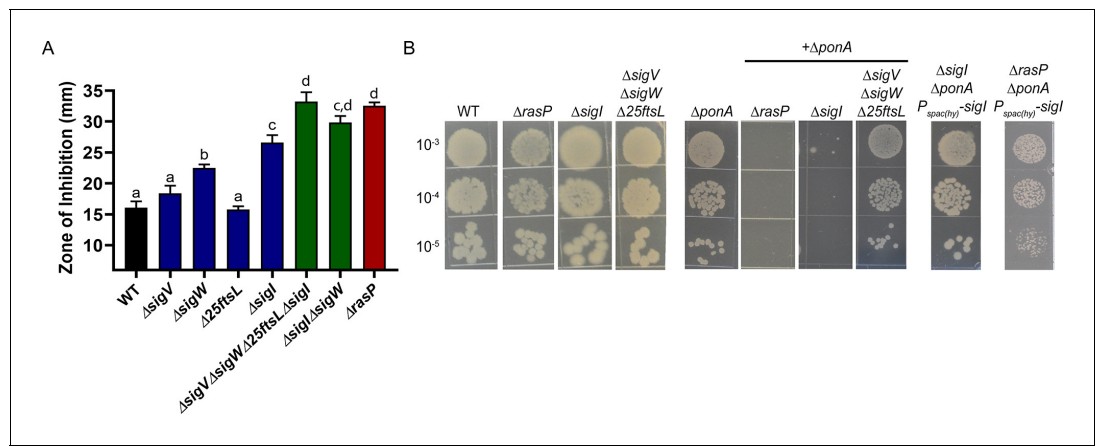

**Figure 3.** The EcsAB-RasP pathway functions largely through *sigI*. (A) CEF (10 µg) sensitivity (disc diffusion assay) for WT, Δ*rasP*, Δ*sigV*, Δ*sigW*, Δ*25ftsL*, Δ*sigI*, Δ*sigW*Δ*sigI* and Δ*sigV*Δ*sigW*Δ*25ftsL*Δ*sigI* strains. P-value cut-off of <0.0001 was used. (B) Plating efficiency of Δ*rasP*, Δ*sigI* and Δ*sigV*Δ*sigW*Δ*25ftsL* strains in WT and Δ*ponA* deletion background. This assay was done by plating 10 µL of mid-log phase cultures (grown in LB with 20 mM MgSO₄) on LB agar plates (no Mg supplementation). The plating efficiency of Δ*sigI*Δ*ponA* double mutant was also evaluated after ectopic expression of *sigI* from the leaky promoter P_{spac(hy)}.

The online version of this article includes the following source data and figure supplement(s) for figure 3:

**Source data 1.** Data of zone of inhibition.

**Figure supplement 1.** σ^I and RasP have similar MIC against MOE.

**Figure supplement 2.** RasP functions primarily through σ^I to provide resistance against CEF.

functions in undecaprenylpyrophosphate recycling (*Bernard et al., 2005*; *Zhao et al., 2016*; *Radeck et al., 2017b*), and MreBH and LytE are both elongasome-associated proteins. MreBH, one of three MreB-family proteins that associate with the elongasome, sequesters and directs the LytE endopeptidase to the sites of insertion of new peptidoglycan (*Carballido-López et al., 2006*). To further define the role of σ[I] in sustaining viability during aPBP inhibition, we conducted CEF/MOE sensitivity assays using single mutants of σ[I]-controlled genes. The *mreBH*, *lytE* and *bcrC* single mutants exhibited slightly higher sensitivity for both CEF and MOE (*Figure 4—figure supplement 1*), however, they did not entirely phenocopy the *sigI* phenotype. The Δ*mreBH*Δ*lytE* double mutant exhibited the same level of CEF and MOE sensitivity as both the *rasP* and *sigI* mutants (*Figure 4A–*

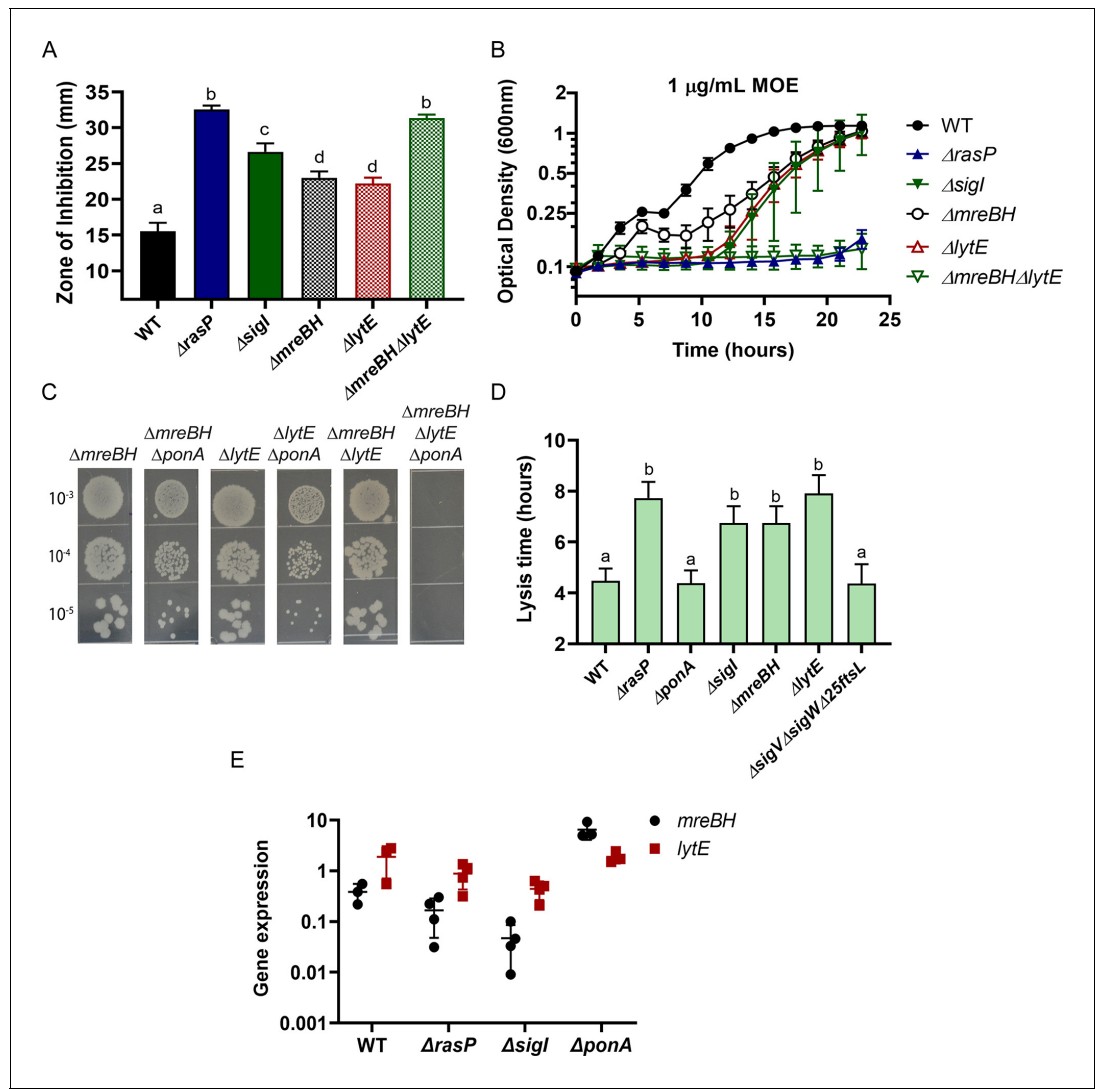

**Figure 4.** σ[I] functions by increasing expression of *mreBH* and *lytE*. (**A**) CEF (10 µg) sensitivity (disc diffusion assay) of Δ*mreBH*, Δ*lytE* and Δ*mreBH*Δ*lytE* strains. Significance was determined with a P-value cut-off of <0.0001. (**B**) Growth kinetics of the mutants in LB medium with 1 µg/mL MOE. (**C**) Plating efficiency of the Δ*mreBH*, Δ*lytE,* and Δ*mreBH*Δ*lytE* mutants alone and in combination with Δ*ponA*. (**D**) The autolytic potential of the cells (WT, Δ*ponA*, Δ*rasP*, Δ*sigI*, Δ*mreBH*, Δ*lytE* and Δ*sigV*Δ*sigW*Δ*25ftsL*) measured by the time taken to reach 50% of initial cell density on treatment with sodium azide. P-value cut-off of <0.0001 was used. (**E**) Gene expression values ($2^{-\Delta ct}$) of *mreBH* and *lytE* normalized to *gyrA* plotted on $\log_{10}$ scale for WT, Δ*rasP*, Δ*sigI* and Δ*ponA* strains.

The online version of this article includes the following source data and figure supplement(s) for figure 4:

**Source data 1.** Data of zone of inhibition, MOE growth kinetics, lysis time and gene expression.

**Figure supplement 1.** σ[I] regulates the expression of *mreBH* and *lytE* to support elongasome function.

*B*). Thus, these results suggest that the EcsAB-RasP-σ$^I$ pathway primarily acts through MreBH and LytE to control elongasome function.

To further validate the importance of MreBH and LytE, we created deletion mutants in the Δ*ponA* background (*Figure 4C*). A Δ*mreBH*Δ*ponA* double mutant could be constructed only when the cells were initially plated on LB supplemented with high Mg$^{2+}$. Once constructed, however, this mutant and the Δ*lytE*Δ*ponA* double mutant did not exhibit a plating defect on LB. In contrast, the triple mutant of Δ*mreBH*Δ*lytE*Δ*ponA* was synthetic lethal and could not be plated on LB agar without Mg$^{2+}$ supplementation. These data suggest an additive role for MreBH and LytE in the effective functioning of the elongasome, likely due to the ability of LytE to retain some function in the absence of MreBH, and MreBH having functional roles beyond localization of LytE.

*B. subtilis* has two partially redundant D,L-endopeptidases, LytE and CwlO, which are collectively essential for cell viability (*Hashimoto et al., 2012*). The involvement of σ$^I$ in the expression of *lytE* has already been established since both Δ*lytE*Δ*cwlO* and Δ*sigI*Δ*cwlO* are synthetic lethal (*Salzberg et al., 2013*). Consistently, Δ*rasP*Δ*cwlO* was also synthetic lethal (*Figure 4—figure supplement 1*). To confirm that LytE activity was reduced in the *rasP* and *sigI* mutants we evaluated the autolytic potential of the cells. Cells were treated with sodium azide, which disrupts membrane potential and activates autolysins (*Jolliffe et al., 1981*; *Wang et al., 2014*). By monitoring the time taken for a 50% reduction in optical density, we found that the Δ*lytE* mutant had a lower rate of autolysis (*Figure 4D*). Similar to Δ*lytE*, we observed that Δ*rasP*, Δ*sigI* and Δ*mreBH* also had lower autolytic potential, consistent with a role in affecting LytE expression or activity.

Next, we evaluated the expression levels of *mreBH* and *lytE* in Δ*rasP*, Δ*sigI* and Δ*ponA* mutants (*Figure 4E*). In the Δ*ponA* mutant, *mreBH* was significantly upregulated, whereas *lytE* was unchanged. In Δ*sigI*, both *mreBH* and *lytE* expression was significantly lower. This suggests that Δ*ponA* cells require higher levels of MreBH to direct the autolytic activity of LytE to support optimal elongasome function, and that activation of σ$^I$ mediates increased *mreBH* expression. As a result, the reduced expression of *mreBH* in Δ*rasP* and Δ*sigI* strains likely contributes to the synthetic lethality with Δ*ponA*.

## Balance in the MreBH-LytE activity is essential for optimal elongasome function

We complemented the conditional essentiality of *mreBH* and *lytE* by ectopically expressing each of these genes individually as well as in combination in different mutant backgrounds. These strains were used to evaluate the relative importance of each gene upon inhibition of PBP1 by monitoring their CEF resistance. Although ectopic expression of *mreBH* complements the CEF sensitivity of Δ*mreBH*, it is unable to restore CEF resistance to the Δ*mreBH*Δ*lytE* double mutant (*Figure 5A*). However, when both *mreBH* and *lytE* were ectopically expressed, the strain was significantly more CEF resistant than WT (*Figure 5A*). Similarly, induction of *mreBH* modestly increased CEF resistance of Δ*rasP* (*Figure 5B*), but not a Δ*rasP*Δ*lytE* double mutant. Similar results were obtained in cells where *pbpD*, *pbpF* and *pbpG* were deleted (data not shown) indicating no indirect effect of MreBH on these aPBPs. In Δ*sigI*, however, *mreBH* expression alone had no significant impact on CEF resistance, perhaps due to reduced availability of LytE. Thus, increasing MreBH levels likely functions to increase elongasome activity by facilitating the localized action of LytE. Conversely, the P$_{spac(hy)}$*lytE* overexpression construct could not be introduced into the Δ*rasP* and Δ*sigI* mutants. We speculate that high LytE, in cells that have reduced expression of *mreBH*, leads to delocalized and unregulated autolysin activity. Collectively, these results further support a model in which a major role of MreBH is in directing LytE to sites of ongoing, elongasome-dependent PG synthesis.

The elongasome is critical for the maintenance of rod-shape, as judged by the spherical morphology of conditional mutants that are depleted for either the RodA transglycosylase or the two class B PBPs that provide transpeptidase activity (*Boylan and Mendelson, 1969*; *Wei et al., 2003*). The maintenance of rod shape is also affected by the balance between the directional motion of the elongasome and the random diffusive motion of PBP1 (*Dion et al., 2019*). Any imbalance in the activities of the two systems can lead to change in cell morphology. Overexpression of MreB or other elongasome proteins leads to cells that are longer and thinner, whereas overexpression of PBP1 leads to shorter and wider cells (*Dion et al., 2019*). Thus, we hypothesized that the effects of

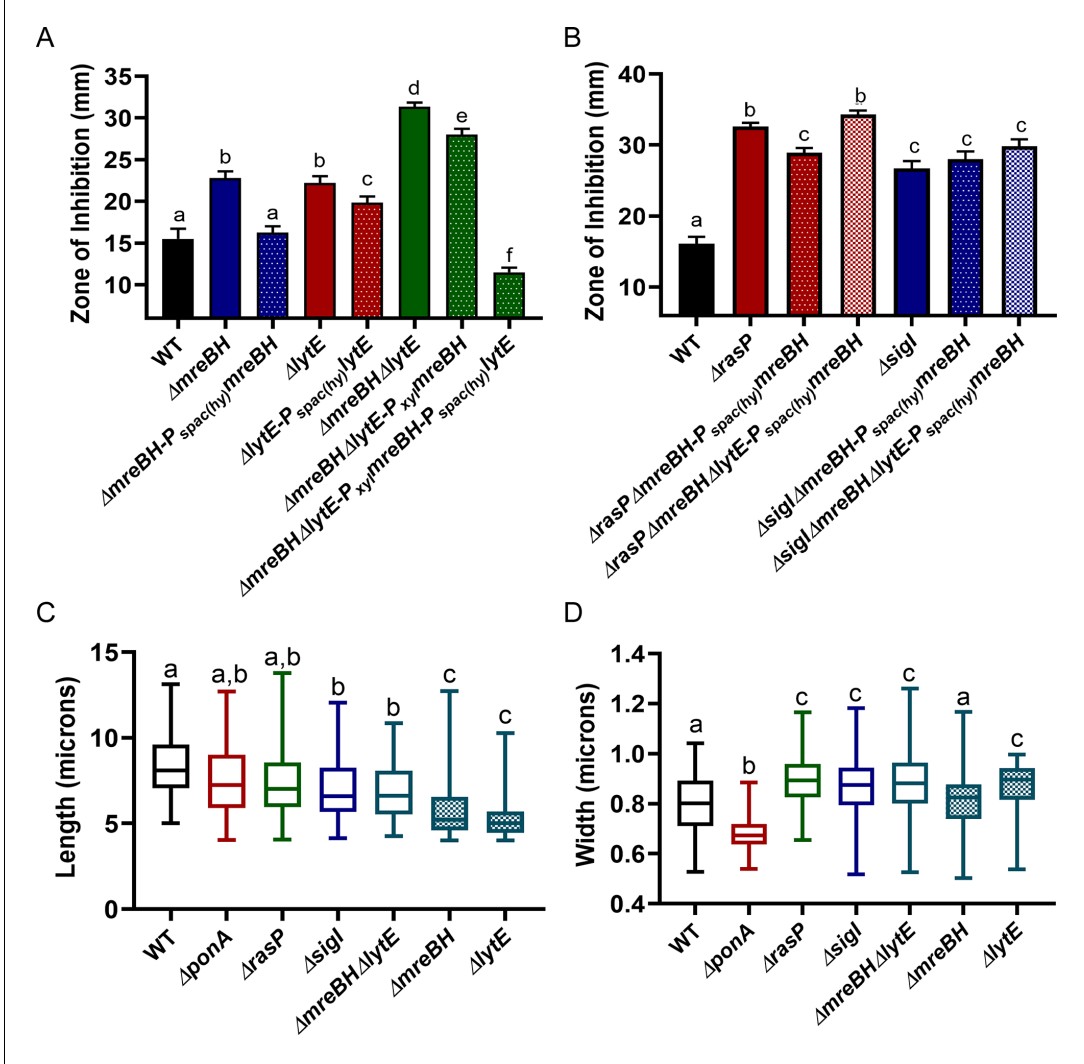

**Figure 5.** MreBH and LytE function cooperatively to increase elongasome function. (**A**) CEF (10 μg) sensitivity (disc diffusion assay) of the Δ*mreBH*, Δ*lytE,* and Δ*mreBH*Δ*lytE* strains with and complementation by ectopic expression of genes from the leaky promoter, P$_{spac(hy)}$, or (for the Δ*mreBH*Δ*lytE* strain) expression of *mreBH* from a xylose inducible promoter (P$_{xyl}$) and *lytE* from the P$_{spac(hy)}$. P-value cut-off of <0.0001 was used. (**B**) CEF sensitivity (as for panel A) for Δ*rasP* and Δ*sigI* mutants with ectopic expression of *mreBH* from P$_{spac(hy)}$ in the presence and absence of *lytE*. P-value cut-off of <0.0001 was used. Cell length (**C**) and width (**D**) of WT, Δ*ponA*, Δ*rasP*, Δ*sigI*, Δ*mreBH*Δ*lytE*, and Δ*mreBH* and Δ*lytE* strains was determined using at least 100 cells for each strain. P-value cut-off of <0.005 was used.

The online version of this article includes the following source data for figure 5:

**Source data 1.** Data of zone of inhibition and cell size measurements.

the σ$^I$ regulatory system (acting through *mreBH* and *lytE*) on elongasome function would be revealed by monitoring cell morphology. We imaged WT, Δ*rasP*, Δ*sigI*, Δ*mreBH*, Δ*lytE*, Δ*mreBH*Δ*lytE* and Δ*ponA* cells and quantified the cell length and width using MicrobeJ (*Ducret et al., 2016*). Indeed, Δ*rasP*, Δ*sigI* and Δ*mreBH*Δ*lytE* mutants were significantly shorter (*Figure 5C*) and wider (*Figure 5D*) compared to the WT, which indicates that these cells were primarily utilizing PBP1 for PG synthesis. Δ*mreBH* and Δ*lytE* mutants individually also had lower elongasome activity. In contrast, the Δ*ponA* mutant formed significantly thinner cells due to PG synthesis being contributed mainly by the elongasome. These data support the conclusion that the *rasP*, *sigI* and *mreBH-lytE* genes all support elongasome function.

## Suppressor analysis confirms the importance of *mreBH* and *lytE* in cells dependent on elongasome

Next, we took advantage of the Δ*rasP*Δ*ponA* synthetic lethality to isolate suppressors that grow on LB agar plates. Using whole-genome resequencing, we identified three strains with point mutations in *walK* (Ala241Asp, Ser385Leu, Asp274Ala). WalK is the sensor kinase of the essential two-component system WalKR, which regulates cell wall metabolism (*Takada and Yoshikawa, 2018*). WalR has binding sites upstream of *sigI*, *mreBH* and *lytE* and activates expression of these genes under heat stress (*Huang et al., 2013*). In addition to their regulation by σ$^I$, *sigI* and *lytE* also have σ$^A$-dependent promoters. WalR may function in conjunction with the σ$^A$ holoenzyme, which together with σ$^I$ controls *lytE* expression (*Tseng et al., 2011*). Taking into account the importance of WalKR in the expression of *sigI*, *mreBH* and *lytE*, we characterized one of the suppressor mutants of WalK, wherein aspartate 274 is changed to alanine (D274A).

Residue 274 lies in the cytoplasmic Per-Arnt-Sim (PAS) domain of WalK (*Figure 6A*). PAS domains have been linked to signal sensing (*Taylor and Zhulin, 1999*) and may be involved in protein dimerization (*Huang et al., 1993*). Recently, the cytoplasmic PAS domain of *S. aureus* WalK was found to bind zinc at a site including D274. Moreover, mutation in this binding site, which is highly conserved in WalK orthologs (*Monk et al., 2019*), led to increased kinase activity. We therefore hypothesized that the WalK$^{D274A}$ suppressor (denoted as WalK*) led to increased activity of the WalKR two-component system. We used CRISPR mutagenesis to introduce the *walK** allele into WT cells and then confirmed that this allele suppressed the synthetic lethality of Δ*rasP*Δ*ponA* (*Figure 6B*).

We next aimed to test the effect of WalK* on gene expression and cell wall homeostasis. The *sigI* and *lytE* genes can be expressed through their σ$^A$ promoter after activation by WalR (*Salzberg et al., 2013*; *Tseng et al., 2011*). However, *mreBH* lacks an annotated σ$^A$ promoter, implying that the expression of *mreBH* may rely on WalR activation of the σ$^I$ holoenzyme. To test

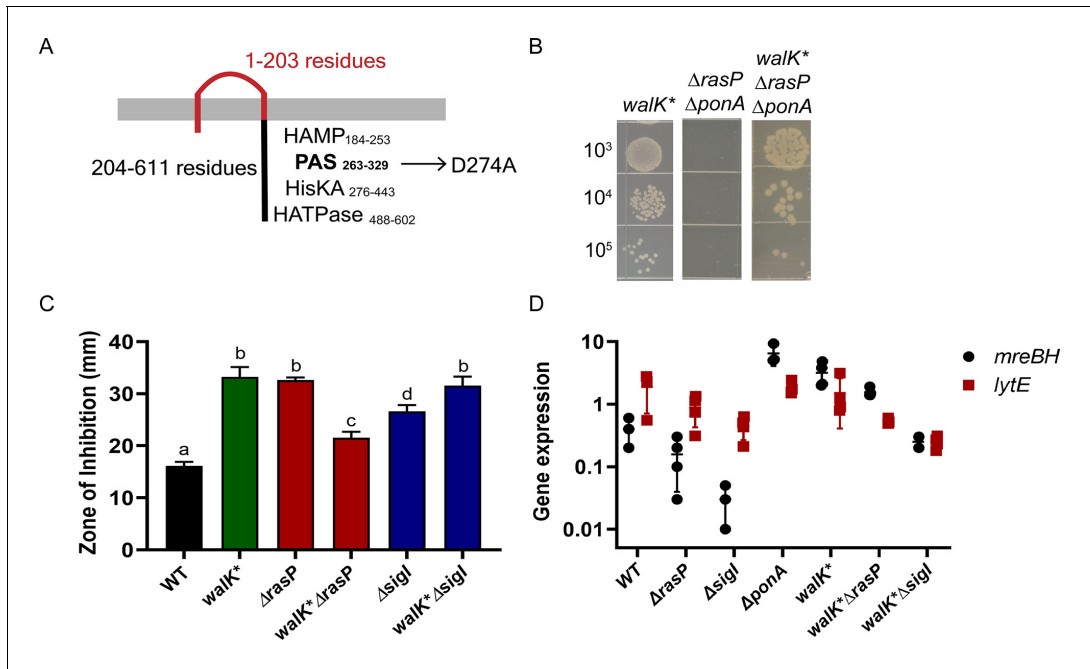

**Figure 6.** A *walK** suppressor mutation elevates *mreBH* transcription. (**A**) The D274 residue of WalK is part of a PAS-domain associated Zn-binding motif. (**B**) A *walK** mutation rescues growth of the Δ*rasP*Δ*ponA* strain as monitored by a spot dilution assay. (**C**) CEF (10 μg) resistance (disc diffusion assay) of Δ*rasP* and Δ*sigI* and the respective double mutants of *walK**Δ*rasP* and *walK**sigI*. A P-value cut-off of <0.0001 was used. (**D**) The effect of *walK** on the expression profile of *mreBH* and *lytE* genes, alone and in combination with Δ*rasP* and Δ*sigI*. The gene expression values (2$^{-\Delta ct}$) were normalized with the house-keeping gene *gyrA* and then plotted on a log$_{10}$ scale.

The online version of this article includes the following source data for figure 6:

**Source data 1.** Data of zone of inhibition and gene expression.

this hypothesis, we measured CEF sensitivity of *walK\*ΔrasP* and *walK\*ΔsigI* strains (**Figure 6C**). Although *walK\** increased CEF resistance of the *ΔrasP* strain, it could not rescue the *ΔsigI* strain. This supports the idea that WalR may act in conjunction with σ$^I$ to activate transcription of *mreBH*, and thereby augment elongasome activity. Increased activation of WalK\* can lead to increased expression of not only *lytE*, but also *cwlO* (**Takada and Yoshikawa, 2018**). This could lead to elevated autolysin levels that might account for the higher CEF sensitivity of *walK\** alone compared to WT.

We further quantified the mRNA levels of *mreBH* and *lytE* in the *walK\** strain and in the *walK\*ΔrasP* and *walK\*ΔsigI* strains (**Figure 6D**). The *walK\** allele led to increased expression of both *mreBH* and *lytE*. Moreover, these levels were similar to that observed in the *ΔponA* background, suggesting that deletion of *ponA* leads to a compensatory increase in *mreBH* and *lytE* mediated by the WalKR. However, they were lower for the *walK\*ΔsigI* strain. These data suggest that *walK\** leads to increased activation of WalR, which then leads to increased transcription of *sigI* and thereby of *mreBH* and *lytE*. This ultimately leads to the survival of the *ΔrasPΔponA* strain. These data also validate the importance of RasP and σ$^I$ in the regulation of MreBH and LytE and their significant impact on elongasome activity, especially in the *ΔponA* background.

## Additive role of σ$^I$ and σ$^M$ in regulating the elongasome activity

While our results suggest a critical role for σ$^I$ in aPBP-elongasome homeostasis through its regulation of MreBH and LytE, previous studies have indicated that the extracytoplasmic (ECF) sigma factor σ$^M$ also plays a significant role in *B. subtilis* cell wall homeostasis. σ$^M$ regulates the expression of *rodA*, *mreB*, *mreC* and *mreD* (core components of the elongasome), as well as *ponA* and other genes involved in PG synthesis (**Eiamphungporn and Helmann, 2008**; **Luo and Helmann, 2012**). To determine the relative contribution of σ$^M$ to cell survival during aPBP inhibition, we used $P_M$\* mutations that selectively inactivate σ$^M$-dependent promoters of genes encoding elongasome components. We constructed the $P_M$\**rodA* and $P_M$\**ponA* strains that are unable to upregulate *rodA* and *ponA*, respectively, and a $P_M$\**maf* strain that cannot upregulate the *mreBCD* genes located downstream of the intragenic $P_M$ inside *maf* (**Eiamphungporn and Helmann, 2008**). We also constructed the double mutant $P_M$\**rodA* $P_M$\**maf* strain. The CEF sensitivity of $P_M$\**rodA* and $P_M$\**rodA*-$P_M$\**maf* was similar to that of the *sigM* mutant (**Figure 7A**). Neither $P_M$\**maf* nor $P_M$\**ponA* were CEF sensitive. Thus, under conditions where CEF has inhibited PBP1, σ$^M$ helps restore peptidoglycan synthesis primarily by increasing the expression of *rodA* to increase elongasome activity. In contrast, the double mutants of *ΔecsAΔsigM*, *ΔrasPΔsigM* and *ΔsigIΔsigM* revealed an additive effect with respect to CEF

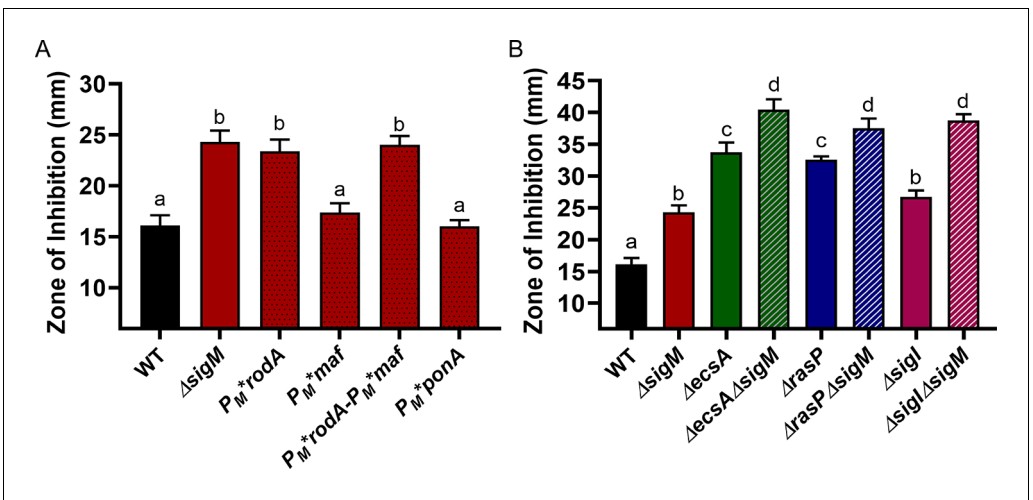

**Figure 7.** σ$^M$ contributes additively with σ$^I$ to CEF resistance by increasing expression of *rodA*. CEF (10 μg) sensitivity (disc diffusion assay) for (**A**) WT, *ΔsigM* and promoter mutants of $P_M$\**rodA*, $P_M$\**maf* (which controls expression of *mreBCD*), $P_M$\**rodA*-$P_M$\**maf* and $P_M$\**ponA* and (**B**) WT and *ΔsigM* mutants, alone and in combination with *ΔecsA, ΔrasP* and *ΔsigI*. P-value cut-off of <0.0001 was used for both the graphs.

The online version of this article includes the following source data for figure 7:

**Source data 1.** Data of zone of inhibition.

sensitivity (*Figure 7B*). Thus, the role of the elongasome in PG synthesis can be regulated through two-independent pathways: the EcsAB-RasP-σ$^I$ pathway acts by regulating MreBH and LytE, and the σ$^M$ pathway acts through RodA.

## Discussion

Peptidoglycan (PG) is a defining feature of bacteria. This cellular enclosure must provide stability, yet at the same time be highly dynamic and adaptable. During growth, PG is continuously remodeled, which involves the action of autolysins, hydrolytic enzymes that cleave links within and between the glycan strands (*Vollmer et al., 2008*; *Egan et al., 2020*). These hydrolases are essential for the insertion of new glycan strands into the existing structure (*Hashimoto et al., 2012*; *Singh et al., 2012*). Cell shape maintenance requires that the sites of new PG synthesis be spatially regulated, often in response to the activity of cytoskeletal filaments such as the MreB (*Domínguez-Escobar et al., 2011*) and FtsZ proteins (*Mahone and Goley, 2020*).

*B. subtilis*, a genetically tractable model organism, has provided an important system for investigating the pathways of PG synthesis in rod-shaped, Gram positive bacteria. During cell elongation, a multiprotein complex designated the elongasome is the primary biosynthetic machine for inserting new glycan strands. In *B. subtilis*, there are three MreB paralogs (MreB, Mbl and MreBH), which colocalize to form elongasome-associated cytoskeletal filaments along the cell periphery (*Carballido-López et al., 2006*; *Garner et al., 2011*). Cells lacking all three paralogs lose their rod shape and become spheres which ultimately lyse (*Kawai et al., 2009*). Whereas MreB and Mbl are critical for the circumferential motion of the elongasome, the role of MreBH is less clear, and seems related to its ability to recruit LytE (*Carballido-López et al., 2006*). PG synthesis by the elongasome relies on the activity of RodA as TG, with bPBPs providing TP activity (*Figure 8A*). A separate complex, the divisome, builds the cross-walls prior to cell separation (*Mahone and Goley, 2020*).

Because of its unique chemical composition, PG synthesis requires numerous highly conserved enzymes, which thereby present attractive targets for antibiotics (*Bugg et al., 2011*). Inhibitors of PG synthesis may result in spheroplast formation, cell lysis, or morphological defects, depending on the antibiotic target and the organism (*Cross et al., 2019*; *Emami et al., 2017*). Many of our most familiar antibiotics are natural products of soil bacteria, including *Bacillus* spp. (*Kaspar et al., 2019*; *Stein, 2005*) and many actinobacteria (*Mahajan, 2012*). Like other soil bacteria, *B. subtilis* has

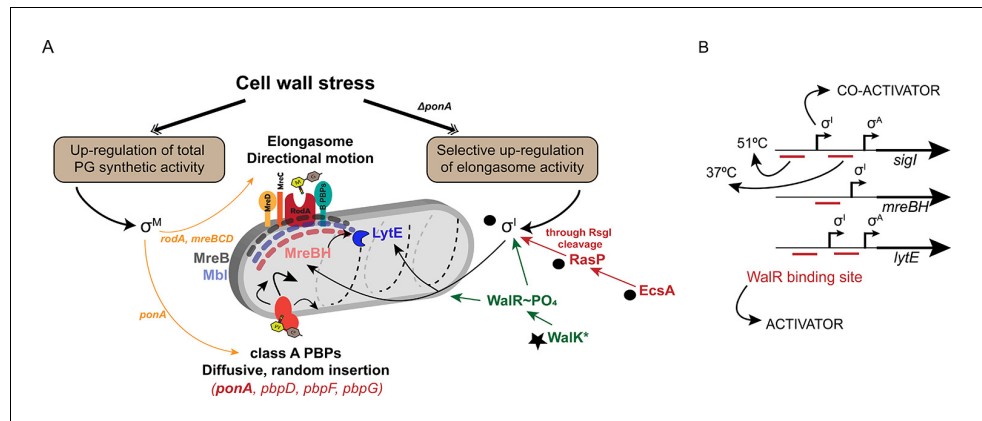

**Figure 8.** σ$^I$ co-ordinates with WalKR to regulate elongasome function, and complements the σ$^M$ dependent stress response. (**A**) PG synthesis potential is dictated by the activity of the elongasome in collaboration with aPBPs. Cell wall stress activates σ$^M$ (left), which up-regulates both pathways. In the absence of aPBPs, cells up-regulate elongasome activity through σ$^I$, which increases expression of genes (*mreBH* and *lytE*) important for elongasome function. Synthetic lethal relationships are shown here between deletion of *ponA* and genes in the σ$^I$ pathway (black circles). Bypass of synthetic lethality can be compensated by a gain of function mutation in *walK* (star). (**B**) The promoter regions of *sigI*, *mreBH* and *lytE* are shown, depicting the binding sites of WalR and σ$^I$ as annotated before (*Huang et al., 2013*). σ$^I$ and WalR act as activators for the expression of *sigI* and *lytE* from the σ$^A$ promoter. The downstream WalR binding site is important for expression of *sigI* and *lytE* at 37°C whereas the upstream binding site is crucial for the heat induction of these genes at 51°C.

substantial intrinsic resistance to many antibiotics (*Kingston et al., 2013*; *Radeck et al., 2017a*; *Helmann, 2016*). We have explored these intrinsic resistance mechanisms by analysis of cell envelope stress responses, including those controlled by alternative sigma factors (*Helmann, 2016*). For example, $\sigma^V$ is induced by and provides resistance to lysozyme by covalently modifying PG (*Guariglia-Oropeza and Helmann, 2011*), whereas $\sigma^W$ is induced by and provides resistance to membrane-active bacteriocins (*Butcher and Helmann, 2006*; *Kingston et al., 2011*).

The $\sigma^M$ response is selectively induced by stresses during PG synthesis and contributes to resistance to a wide-variety of PG synthesis inhibitors, including MOE, CEF, and bacitracin (*Helmann, 2016*; *Mascher et al., 2007*). The $\sigma^M$ regulon serves to both upregulate PG synthetic capacity, and to compensate for stresses resulting from PG inhibition. The former includes the up-regulation of elongasome components (*Figure 8A*) and PG biosynthetic enzymes (PBP1, Ddl, MurB, MurF, BcrC, Amj) (*Eiamphungporn and Helmann, 2008*). The latter includes the large regulon controlled by the Spx transcription factor that protects cells against antibiotic-associated oxidative stress (*Rojas-Tapias and Helmann, 2018*). Finally, it has recently been shown that induction of a $\sigma^M$-regulated ppGpp synthase, YwaC, increases the number of persister cells following antibiotic exposure (*Fung et al., 2020*).

Here, we identify a major role for another alternative sigma factor, $\sigma^I$, in conferring intrinsic resistance to important cell wall antibiotics, MOE and CEF. Induction of $\sigma^I$, which requires the EcsAB-RasP regulatory pathway (*Liu et al., 2017*), selectively elevates elongasome function by increasing the expression of the MreB paralog, MreBH, and the associated autolytic endopeptidase LytE (*Carballido-López et al., 2006*). This stress response is critical in cells lacking PBP1, as judged by the synthetic lethality of Δ*sigI* Δ*ponA* mutants (*Figure 3B*). This stress response functions in coordination with both the $\sigma^M$ stress response (*Figure 7A*), which increases elongasome function by upregulation of the RodA TG (*Meeske et al., 2016*; *Emami et al., 2017*), and the essential WalKR two-component system (*Figures 6 and 8*). Although $\sigma^I$ was previously linked to heat-stress (*Zuber et al., 2001*), virulence in *B. anthracis* (*Kim and Wilson, 2016*), and control of autolysin synthesis (*Salzberg et al., 2013*), our results reveal new insights into its role in cell envelope stress.

This study also highlights the complex regulation of the *mreBH* and *lytE* genes. WalR, $\sigma^I$ and $\sigma^A$ binding sites have been previously annotated in the promoters of *sigI*, *mreBH* and *lytE* (*Figure 8B*). The WalK (D274A) gain of function mutant suppresses the lethal phenotype of Δ*rasP*Δ*ponA* by induction of *mreBH* and *lytE* (*Figure 6*). However, induction was not significant in the $\sigma^I$ mutant. We conclude that co-activation by WalR and $\sigma^I$ is required for induction of these two genes. The signals sensed by WalK are unclear, but it was recently suggested that peptidoglycan cleavage products generated by LytE and CwlO can be sensed by WalK to balance the activity of these proteins (*Dobihal et al., 2019*). Moreover, it was previously observed that *sigI* activation enhances the growth of *mbl* mutants (*Schirner and Errington, 2009*), which we suggest was likely due to increasing elongasome activity through *mreBH* and *lytE*.

Collectively, our results reveal that WalKR and $\sigma^I$ act in coordination to maintain optimal elongasome activity, and these pathways complement the general PG stress response activated by $\sigma^M$ (*Figure 8*). One general theme that has emerged is that PG synthesis involves multiple, functionally overlapping systems, often with one being inducible by antibiotic inhibition of the other. For example, the inducible UPP phosphatase BcrC complements the activity of UppP (*Radeck et al., 2017b*; *Zhao et al., 2016*), and the $\sigma^M$-regulated Amj functions as a second lipid II flippase that is critical when MurJ is inhibited (*Chamakura et al., 2017*; *Meeske et al., 2015*). Similarly, inhibition of aPBPs by MOE leads to an essential, compensatory induction of RodA (*Meeske et al., 2016*; *Emami et al., 2017*). Here, it is shown that this single $\sigma^M$-regulated target gene can largely account for the CEF sensitivity of *sigM* mutants (*Figure 7*). This increase in RodA, together with the induction of MreBH and LytE, serves to boost the biosynthetic potential of the elongasome. These results reveal mechanisms that allow diverse PG biosynthetic complexes to coordinate their activities, in both time and space. The highly orchestrated processes that direct and coordinate PG synthesis are important both for intrinsic antibiotic resistance, as explored here and are ultimately responsible for the enormous diversity of bacterial morphologies (*Caccamo and Brun, 2018*).

# Materials and methods

## Key resources table

| Reagent type (species) or resource | Designation | Source or reference | Identifiers | Additional information |
|---|---|---|---|---|
| Strain, strain background (*Bacillus subtilis*, strain 168) | WT | Lab stock | *B. subtilis* 168 | (see Materials and methods) |
| Recombinant DNA reagent | | This study | *E. coli* with pMarA1 | (see Materials and methods) |
| Recombinant DNA reagent | HB20725 | This study | 168 pMarA1 | (see Materials and methods) |
| Recombinant DNA reagent | HB20738 | This study | pbpDFG null; ponA::erm;pMarA | (see Materials and methods) |
| Strain, strain background (*Bacillus subtilis*, strain 168) | Δ4 Class A PBP | This study | ponA::erm; pbpDFG::null | (see Materials and methods) |
| Strain, strain background (*Bacillus subtilis*, strain 168) | ponA::erm $P_{spank*}$-ponA | This study | ycgO::$P_{spank*}$-ponA; ponA::erm | (see Materials and methods) |
| Strain, strain background (*Bacillus subtilis*, strain 168) | pbpDFG ponA::erm $P_{spank*}$-ponA | This study | pbpDFG::null; ycgO::$P_{spank*}$-ponA; ponA::erm | (see Materials and methods) |
| Strain, strain background (*Bacillus subtilis*, strain 168) | ecsA ponA::erm $P_{spank*}$-ponA | This study | ecsA::null; ycgO::$P_{spank*}$-ponA; ponA::erm | (see Materials and methods) |
| Strain, strain background (*Bacillus subtilis*, strain 168) | pbpDFG ecsA- ponA::erm $P_{spank*}$-ponA | This study | ecsA::null;pbpDFG::null; ycgO::$P_{spank*}$-ponA; ponA::erm | (see Materials and methods) |
| Strain, strain background (*Bacillus subtilis*, strain 168) | ytxG ponA::erm $P_{spank*}$-ponA | This study | ytxG::null; ycgO::$P_{spank*}$-ponA; ponA::erm | (see Materials and methods) |
| Strain, strain background (*Bacillus subtilis*, strain 168) | *pbpDFG ytxG* ponA::erm $P_{spank*}$-ponA | This study | ytxG::null;pbpDFG::null; ycgO::$P_{spank*}$-ponA; ponA::erm | (see Materials and methods) |
| Strain, strain background (*Bacillus subtilis*, strain 168) | ΔecsA | This study | ecsA::kan | (see Materials and methods) |
| Strain, strain background (*Bacillus subtilis*, strain 168) | ΔrasP | This study | rasP::kan | (see Materials and methods) |
| Strain, strain background (*Bacillus subtilis*, strain 168) | ΔponA | This study | ponA::erm | (see Materials and methods) |
| Strain, strain background (*Bacillus subtilis*, strain 168) | ΔecsAΔponA | This study | ecsA::null;ponA::erm | (see Materials and methods) |

*Continued on next page*

*Continued*

| Reagent type (species) or resource | Designation | Source or reference | Identifiers | Additional information |
|---|---|---|---|---|
| Strain, strain background (*Bacillus subtilis*, strain 168) | ΔrasPΔponA | This study | rasP::null;ponA::erm | (see Materials and methods) |
| Strain, strain background (*Bacillus subtilis*, strain 168) | ΔecsAΔrasP | This study | ecsA::null;rasP::erm | (see Materials and methods) |
| Strain, strain background (*Bacillus subtilis*, strain 168) | ΔecsA P$_{spac(hy)}$-ecsA | This study | amyE::P$_{spac(hy)}$-ecsA; ecsA::erm | (see Materials and methods) |
| Strain, strain background (*Bacillus subtilis*, strain 168) | ΔecsA P$_{spac(hy)-}$ecsAecsB | This study | amyE::P$_{spac(hy)}$-ecsAB; ecsA::erm | (see Materials and methods) |
| Strain, strain background (*Bacillus subtilis*, strain 168) | ΔrasP P$_{spac(hy)}$-rasP | This study | amyE::P$_{spac(hy)}$-rasP; rasP::erm | (see Materials and methods) |
| Strain, strain background (*Bacillus subtilis*, strain 168) | ΔsigW | This study | sigW::null | (see Materials and methods) |
| Strain, strain background (*Bacillus subtilis*, strain 168) | ΔsigV | This study | sigV::null | (see Materials and methods) |
| Strain, strain background (*Bacillus subtilis*, strain 168) | ΔsigI | This study | sigI::null | (see Materials and methods) |
| Strain, strain background (*Bacillus subtilis*, strain 168) | Δ25ftsL | This study | Made using CRISPR to remove the 2-26th AAs of FtsL so it is no longer a target of RasP | (see Materials and methods) |
| Strain, strain background (*Bacillus subtilis*, strain 168) | ΔsigVΔsigW Δ25ftsLΔsigI | This study | sigV::null;sigW::null; Δ25ftsL;sigI::kan | (see Materials and methods) |
| Strain, strain background (*Bacillus subtilis*, strain 168) | ΔsigIΔsigW | This study | sigI::null;sigW::kan | (see Materials and methods) |
| Strain, strain background (*Bacillus subtilis*, strain 168) | ΔsigVΔsigW Δ25ftsL | This study | sigV::null;sigW::null; Δ25ftsL | (see Materials and methods) |
| Strain, strain background (*Bacillus subtilis*, strain 168) | ΔsigIΔponA P$_{spac(hy)}$-sigI | This study | sigI::null; amyE:: P$_{spac(hy)}$-sigI; ponA::erm | (see Materials and methods) |
| Strain, strain background (*Bacillus subtilis*, strain 168) | ΔecsAΔsigI | This study | sigI::null;ecsA::kan | (see Materials and methods) |

*Continued on next page*

*Continued*

| Reagent type (species) or resource | Designation | Source or reference | Identifiers | Additional information |
|---|---|---|---|---|
| Strain, strain background (*Bacillus subtilis*, strain 168) | ΔecsAΔsigW | This study | sigW::null;ecsA::kan | (see Materials and methods) |
| Strain, strain background (*Bacillus subtilis*, strain 168) | ΔrasPΔsigI | This study | sigI::null;rasP::kan | (see Materials and methods) |
| Strain, strain background (*Bacillus subtilis*, strain 168) | ΔrasPΔsigW | This study | sigW::null;rasP::kan | (see Materials and methods) |
| Strain, strain background (*Bacillus subtilis*, strain 168) | ΔrsgI | This study | rsgI::null | (see Materials and methods) |
| Strain, strain background (*Bacillus subtilis*, strain 168) | ΔrsiW | This study | rsiW::mls | (see Materials and methods) |
| Strain, strain background (*Bacillus subtilis*, strain 168) | ΔecsAΔrsgI | This study | rsgI::null;ecsA::kan | (see Materials and methods) |
| Strain, strain background (*Bacillus subtilis*, strain 168) | ΔecsAΔrsiW | This study | rsiW::mls;ecsA::kan | (see Materials and methods) |
| Strain, strain background (*Bacillus subtilis*, strain 168) | ΔrasPΔrsgI | This study | rsgI::null;rasP::kan | (see Materials and methods) |
| Strain, strain background (*Bacillus subtilis*, strain 168) | ΔrasPΔrsiW | This study | rsiW::mls;rasP::kan | (see Materials and methods) |
| Strain, strain background (*Bacillus subtilis*, strain 168) | ΔsigM | This study | sigM::null | (see Materials and methods) |
| Strain, strain background (*Bacillus subtilis*, strain 168) | ΔecsAΔsigM | This study | sigM::null;ecsA::kan | (see Materials and methods) |
| Strain, strain background (*Bacillus subtilis*, strain 168) | ΔrasPΔsigM | This study | sigM::null;rasP::kan | (see Materials and methods) |
| Strain, strain background (*Bacillus subtilis*, strain 168) | ΔsigIΔsigM | This study | sigM::null;sigI::kan | (see Materials and methods) |
| Strain, strain background (*Bacillus subtilis*, strain 168) | Pm*rodA | *Zhao et al., 2019* | WT 168 transformed with CRISPR plasmid to remove Pm of rodA | (see Materials and methods) |
| Strain, strain background (*Bacillus subtilis*, strain 168) | Pm* maf | *Zhao et al., 2019* | WT 168 transformed wth pMUTIN to introduce maf-Pm*(TGTT) | (see Materials and methods) |

*Continued on next page*

*Continued*

| Reagent type (species) or resource | Designation | Source or reference | Identifiers | Additional information |
|---|---|---|---|---|
| Strain, strain background (*Bacillus subtilis*, strain 168) | Pm*rodA Pm*murG | This study | Pm*murG transformed with CRISPR plasmid to remove Pm of ProdA | (see Materials and methods) |
| Strain, strain background (*Bacillus subtilis*, strain 168) | Pm*ponA | This study | WT168 transformed with CRISPR plasmid to remove Pm of ponA | (see Materials and methods) |
| Strain, strain background (*Bacillus subtilis*, strain 168) | ΔmreBH | This study | mreBH::null | (see Materials and methods) |
| Strain, strain background (*Bacillus subtilis*, strain 168) | ΔlytE | This study | lytE::null | (see Materials and methods) |
| Strain, strain background (*Bacillus subtilis*, strain 168) | ΔgsiB | This study | gsiB::spec | (see Materials and methods) |
| Strain, strain background (*Bacillus subtilis*, strain 168) | ΔfabI | This study | fabI::null | (see Materials and methods) |
| Strain, strain background (*Bacillus subtilis*, strain 168) | ΔbcrC | This study | bcrC::null | (see Materials and methods) |
| Strain, strain background (*Bacillus subtilis*, strain 168) | ΔmreBHΔlytE | This study | mreBH::null;lytE::null | (see Materials and methods) |
| Strain, strain background (*Bacillus subtilis*, strain 168) | ΔmreBHΔponA | This study | mreBH::null;ponA::erm | (see Materials and methods) |
| Strain, strain background (*Bacillus subtilis*, strain 168) | ΔlytEΔponA | This study | lytE::null;ponA::erm | (see Materials and methods) |
| Strain, strain background (*Bacillus subtilis*, strain 168) | ΔmreBHΔlytE ΔponA | This study | mreBH::null;lytE::null; ponA::erm | (see Materials and methods) |
| Strain, strain background (*Bacillus subtilis*, strain 168) | ΔmreBH $P_{spac(hy)}$-mreBH | This study | mreBH::null; amyE::$P_{spac(hy)}$-mreBH | (see Materials and methods) |
| Strain, strain background (*Bacillus subtilis*, strain 168) | ΔlytE $P_{spac(hy)}$-lytE | This study | lytE::null; amyE::$P_{spac(hy)}$-lytE | (see Materials and methods) |
| Strain, strain background (*Bacillus subtilis*, strain 168) | ΔmreBHΔlytE $P_{xyl}$-mreBH | This study | mreBH::null;lytE::null; lacA::$P_{xyl}$-mreBH | (see Materials and methods) |
| Strain, strain background (*Bacillus subtilis*, strain 168) | ΔmreBHΔlytE $P_{xyl}$-mreBH $P_{spac(hy)}$-lytE | This study | lytE::null; amyE::$P_{spac(hy)}$-lytE; lacA::$P_{xyl}$-mreBH; mreBH::kan | (see Materials and methods) |

*Continued on next page*

*Continued*

| Reagent type (species) or resource | Designation | Source or reference | Identifiers | Additional information |
|---|---|---|---|---|
| Strain, strain background (*Bacillus subtilis*, strain 168) | ΔrasPΔmreBH P$_{spac(hy)}$-mreBH | This study | mreBH::null; amyE::P$_{spac(hy)}$-mreBH; rasP::kan | (see Materials and methods) |
| Strain, strain background (*Bacillus subtilis*, strain 168) | ΔrasPΔmreBH ΔlytE P$_{spac(hy)}$-mreBH | This study | mreBH::null;lytE::null; amyE::P$_{spac(hy)}$-mreBH; rasP::kan | (see Materials and methods) |
| Strain, strain background (*Bacillus subtilis*, strain 168) | ΔsigIΔmreBH P$_{spac(hy)}$-mreBH | This study | mreBH::null; amyE::P$_{spac(hy)}$-mreBH; sigI::kan | (see Materials and methods) |
| Strain, strain background (*Bacillus subtilis*, strain 168) | ΔsigIΔmreBHΔlytE P$_{spac(hy)}$-mreBH | This study | mreBH::null;lytE::null; amyE::P$_{spac(hy)}$-mreBH; sigI::kan | (see Materials and methods) |
| Strain, strain background (*Bacillus subtilis*, strain 168) | walK* | This study | WalK$_{D274A}$, constructed using CRISPR | (see Materials and methods) |
| Strain, strain background (*Bacillus subtilis*, strain 168) | walK*ΔrasP | This study | WalK$_{D274A}$;rasP::kan | (see Materials and methods) |
| Strain, strain background (*Bacillus subtilis*, strain 168) | walK*ΔsigI | This study | WalK$_{D274A}$;sigI::kan | (see Materials and methods) |
| Strain, strain background (*Bacillus subtilis*, strain 168) | walK*ΔrasPΔponA | This study | WalK$_{D274A}$;rasP::kan; ponA::erm | (see Materials and methods) |
| Recombinant DNA reagent | pMarA | Le Breton et al., 2006 | | a plasmid harboring the mariner-Himar1 transposase |
| Recombinant DNA reagent | pMarA1 | | | Modified pMarA to introduce MmeI sites |
| Recombinant DNA reagent | pDR244 | BGSC (ECE274) | | To remove the kan/erm cassette from BKE strains |
| Recombinant DNA reagent | pAM012 | Meeske et al., 2015 | | For Pspank*-ponA constructs |
| Recombinant DNA reagent | pPL82 | | | For Pspac(hy) constructs at amyE locus |
| Recombinant DNA reagent | pBS2EXylRPxylA | BGSC (ECE741) | | For Pxyl constructs at lacA locus |

## Bacterial strains, plasmids and growth conditions

All stains were grown in lysogeny broth (LB) medium at 37°C. Liquid cultures were aerated on an orbital shaker at 300 rpm. Glycerol stocks were streaked on LB agar plates and incubated overnight at 37°C. Conditionally synthetic lethal strains were grown in LB medium with 20 mM MgSO$_4$.

Bacterial strains used in this study have been listed in the Key Resources Table. For all deletion mutants, primary strains were ordered from the BKK/BKE collection available at the Bacillus Genetic Stock Centre (BGSC) (Koo et al., 2017). These gene deletions with the antibiotic cassette (kanamycin or erythromycin) were then transformed into our WT 168 strain using natural competence induced in modified competence (MC) medium. *rasP*, *ecsA* and *ponA* deletion strains had very low natural competence. Thus, other mutations were introduced using SPP1 phage transduction as

described (*Kearns et al., 2005*). The null mutants were constructed using pDR244, which removes the resistance cassette leading to clean in-frame deletions (*Koo et al., 2017*). The resulting gene deletions (designated Δ) were confirmed with check primers listed in *Supplementary file 1*.

Genes were ectopically expressed at *amyE* under promoter $P_{spac(hy)}$ using pPL82 plasmid (*Quisel et al., 2001*). MreBH was also expressed at the *lacA* locus under xylose inducible promoter $P_{xyl}$ using plasmid pECE741 (*Popp et al., 2017*). The respective genes were amplified from genomic DNA using primers listed in *Supplementary file 1*. The required restriction enzyme sites (and if required a ribosome binding site (RBS)) were incorporated in the primers used for gene amplification. CRISPR-Cas9 mutagenesis was carried out using pJOE8999 plasmid as described before (*Altenbuchner, 2016*). The primers used to construct the repair fragment and guide RNAs are in *Supplementary file 1*. The whole sequence of the genes was confirmed by Sanger sequencing (Biotechnology Resources core facility at Cornell University).

## Transposon mutagenesis

The transposon-sequencing (Tn-Seq) was performed using modified pMarA (*Le Breton et al., 2006*). pMarA is a plasmid harboring the mariner-Himar1 transposase gene and a temperature-sensitive replicon to select for transposition events. Two MmeI sites were introduced flanking the BstXI and PstI sites to generate plasmid pMarA1 (HE8334). The plasmid was transformed into WT *Bacillus subtilis* and Δ*pbpDFG ponA::erm* mutant at 28°C selecting for Kan$^R$ on LB plates supplemented with 10 mM MgSO$_4$ (final concentration) to generate strain HB20725 and HB20738, respectively. Liquid cultures of HB20725 and HB20738 harboring plasmid-borne transposons were grown at 28°C in liquid LB medium with 10 mM MgSO$_4$ to mid-exponential phase (OD$_{600}$ ~0.4), diluted and spread on LB plates containing kanamycin and 10 mM MgSO$_4$. Plates were incubated overnight at 48°C to select for transposition events, and the ones with distinct single colonies (not too crowded, and about 500 colonies per plate) were pooled together. Two hundred and forty plates with a total of >100,000 independent colonies were pooled together for each strain, and their genomic DNA was isolated. For each strain, 10 µg of genomic DNA was digested using MmeI, purified and ligated with sequencing adaptors. Illumina sequencing was performed and DNA adjacent to the transposon insertion sites were matched to *Bacillus subtilis* reference genome NC_000964.3 using CLC workbench version 8.5.1. Matching results were visualized using CLC workbench, and quantified using Tn-seq Explorer software (*Solaimanpour et al., 2015*). For visualization of transposon insertions, IGV genome browser was used (*Robinson et al., 2011*).

## Plating efficiency

For plating efficiency (spot dilution) assays, the cultures were grown in LB medium with 20 mM MgSO$_4$ to ~0.4 OD$_{600}$. 1 mL of culture was centrifuged at 5000 rpm for 5 min and resuspended in LB medium (without MgSO$_4$). 10-fold serial dilutions were done in LB medium and 10 µL was plated/spotted on LB agar plates, allowed to air-dry for 10–15 min, and incubated overnight at 37°C.

## Growth kinetics and MIC determinations

Cultures were grown in LB medium to ~0.4 OD$_{600}$. 1 µL of this culture was inoculated in each well containing 200 µL of LB media with the required drug concentration. Honeycomb 100-well plates were used for the assay. The increase in the OD$_{600}$ of the culture was monitored real-time using Bioscreen C growth curve analyzer (Growth curves USA). Readings were taken at every 15 min interval up to 24 hr under constant shaking conditions at 37°C. For MIC determination, two-fold increase in the drug concentration was screened ranging from (0.2 to 1.6 µg/mL). The minimum concentration which inhibited the growth (less than 0.2 OD$_{600}$) up to at least 10 hr of incubation was considered as the MIC for the strain.

## Disc diffusion assays

Antibiotic sensitivity was screened by determining the zone of inhibition using a disc diffusion assay. Cultures were allowed the grow up to ~0.4 OD$_{600}$. 100 µL of this culture was added to 4 mL of top agar (0.75% agar) kept at 50°C to prevent it from solidifying. This was poured on to 15 mL LB agar plates (1.5% agar). The top agar was allowed to air-dry for 30 min. A Whatmann paper filter disc of 6 mm was then put on the top agar. The required amount of drug was added on the disc

immediately. The plates were incubated overnight at 37°C and the diameter of the clear zone of inhibition was measured. For all histograms, the zone of inhibition (Y-axis) starts from 6 mm which is the disc diameter. For strains having the inducible promoter $P_{xyl}$, both the top agar and LB agar plates were made with 0.1% xylose.

## Autolytic potential

200 µL of cells (~0.4 $OD_{600}$) were added in each well of a 100-well honeycomb plate. To this, 0.05 M of sodium azide (from 5 M stock) was added. Immediately, the real-time monitoring of the decrease in $OD_{600}$ was started with Bioscreen C. Readings were taken every 15 min for up to 24 hr. The time at which 50% of the cells had lysed was noted for each mutant. The time taken (in hours) was plotted as lysis time for each strain. Sodium azide stock was prepared fresh before every experiment.

## Real-time PCR

Gene expression for *mreBH* and *lytE* was determined by real-time PCR using primers in Table S2. RNA was purified from 1.5 mL of ~0.4 $OD_{600}$ cells using the RNeasy Kit from Qiagen as per the manufacturer's instructions. 2 µg of RNA was used to prepare 20 µL of cDNA to achieve a final concentration of 100 ng/µL using High capacity cDNA reverse transcription kit from Applied Biosystems. The gene expression levels were measured using 100 ng of cDNA using 0.5 µM of gene specific primers and 1X SYBR green (Bio-Rad) in CFX connect real-time system from Bio-Rad. *gyrA* was used a house-keeping gene. Gene expression values ($2^{-\Delta ct}$) were plotted after normalization with *gyrA*.

## Cell size measurements

A very thin agar pad was prepared on slides from 0.8% agarose. 10 µL of cells (~0.4 $OD_{600}$) were spotted and allowed to air dry for 10 min before putting on a cover slip. Cells were imaged using Olympus BX61. Images were captured using Cooke Sensicam camera system under 100X magnification with immersion oil. The images were then analyzed for their length and width using MicrobeJ (*Ducret et al., 2016*), a plugin for imageJ (*Schneider et al., 2012*).

## Suppressor analysis

Spontaneous suppressors were picked from LB agar plates for *ΔecsAΔponA* and *ΔrasPΔponA*. 12 suppressors were selected from each background and their chromosomal DNA extracted using Qiagen DNA extraction kit. DNA was sequenced using the Illumina platform at the Biotechnology Resources core facility at Cornell University. The results were trimmed, mapped and aligned with the *ΔecsAΔponA* and *ΔrasPΔponA* backgrounds using CLC genomics workbench.

## Statistical analysis

All the experiments were performed with a minimum of 3 biological replicates. For microscopy images, at least 100 cells per strain were quantified for their cell length and width. One-way ANOVA was used to calculate the statistical significance. Tukey's comparison test was used to determine significance between all the strains. P-value cut-offs have been mentioned in the figure legends. Different letters represent data which are significantly different. Same letter represents mean values which are not statistically different. Significance between two strains was determined using student's t-test.

## Acknowledgements

Research reported in this publication was supported by the National Institutes of Health under award number R35GM122461 to JDH. The content is solely the responsibility of the authors and does not necessarily represent the official views of the National Institutes of Health.

We thank Ahmed Gaballa, Gumpanat Mahipant, Daniel Roistacher, Anna Weaver, Ivano Pezzotta, Jessica Willdigg, Chloe Murrell and Annette Choi for their contributions to the early stages of this project. We also thank Alex Meeske and David Rudner for plasmid pAM012 that contains the $P_{spank*}$ promoter.

## Additional information

### Funding

| Funder | Grant reference number | Author |
|---|---|---|
| National Institutes of Health | R35GM122461 | John D Helmann |

The funders had no role in study design, data collection and interpretation, or the decision to submit the work for publication.

### Author contributions

Yesha Patel, Conceptualization, Investigation, Methodology, Writing - original draft, Writing - review and editing; Heng Zhao, Conceptualization, Investigation, Methodology, Writing - original draft; John D Helmann, Conceptualization, Supervision, Funding acquisition, Writing - original draft, Project administration, Writing - review and editing

### Author ORCIDs

Yesha Patel (iD) https://orcid.org/0000-0001-9888-9888
Heng Zhao (iD) https://orcid.org/0000-0002-7322-5513
John D Helmann (iD) https://orcid.org/0000-0002-3832-3249

### Decision letter and Author response

Decision letter https://doi.org/10.7554/eLife.57902.sa1
Author response https://doi.org/10.7554/eLife.57902.sa2

## Additional files

### Supplementary files

• Supplementary file 1. List of primers used in this study.

• Transparent reporting form

### Data availability

All data generated or analysed during this study are included in the manuscript and supporting files.

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
