## [Decision Letter]

**Acceptance summary:**

This study nicely illuminates the finely tuned connections between different aspects of cell wall synthesis and the ability of *Bacillus subtilis* to enhance the activity of one set of enzymes to compensate for defects in another. This flexibility helps ensure the cell envelope remains a robust barrier in the face of diverse environmental stress.

**Decision letter after peer review:**

Thank you for submitting your article "A regulatory pathway that selectively up-regulates elongasome function in the absence of class A PBPs" for consideration by *eLife*. Your article has been reviewed by three peer reviewers, one of whom is a member of our Board of Reviewing Editors, and the evaluation has been overseen by Gisela Storz as the Senior Editor. The following individual involved in review of your submission has agreed to reveal their identity: Petra Levin (Reviewer #1) Ethan C Garner (Reviewer #2).

The reviewers have discussed the reviews with one another and the Reviewing Editor has drafted this decision to help you prepare a revised submission.

Overall, the reviewers thought that the work provides important insight into how *Bacillus subtilis* is able to compensate for loss of class aPBP activity via upregulation of LytE under the control of SigI and WalRK. At the same time, we felt that several conclusions required additional experiments to be fully supported. Specifically:

1) Data indicating that cefuroxime preferentially targets PBP1.

2) Width measurements of single knockout controls of LytE and MreBH, and/or data on how induction (or titration) of one or the other affects width.

Not essential, although both would strengthen the manuscript, are experiments assessing the potential for RasP/WalRK to control minor aPBPs and the impact of *mreBH* and *lytE* expression and their ability to bypass the aPBPs in *ΔrasP*/*ΔsigI* cells.

Equally important, all three reviewers found the manuscript difficult to read, and felt it would be even more so for *eLife* readers who are not familiar with peptidoglycan biosynthesis and cell envelope stress response pathways in *B. subtilis*, or less comfortable with genetic approaches. For convenience, specific reviewer comments related to textual revisions are at the end of this letter. We would like you to pay particular attention to these comments and ensure that the revised manuscript is accessible to the broad audience that reads *eLife*.

1) The authors propose that activation of LytE bypasses the requirement for aPBPs through activation of the elongasome specific TGase, RodA (citing a 2016 Rudner lab paper) and presumably its associated TP enzymes. While exciting if correct, it is unclear specifically how activation of a TGase is sufficient to bypass both the TG and TP activity of the a PBPs and what this means for the growth and physiology of the cell. If there is substantial data explaining how activation of the TGase also leads to activation of its corresponding TPase, this information should be included in the text.

2) “We next sought to identify antibiotics that, unlike MOE, inhibit aPBPs at their TP active site, and can thereby lead to futile cycling.” “This resulted in synergistic inhibition (Figure 2—figure supplement 2), consistent with the same target drug synergy model, as previously described for *E. coli* protein synthesis inhibitors (Yilancioglu, 2019) and drugs used to treat human diseases (Jia et al., 2009). Taken together, our results suggest that CEF preferentially targets PBP1”.

The explanation in this section is incredibly vague and confusing, and it should be clarified. One point specifically, is they invoke the "same target drug synergy model", without definition or explanation. This should be explained better.

Similarly, they introduce this section looking for "antibiotics that, unlike MOE, inhibit aPBPs at their TP active site, and can thereby lead to futile cycling Given they do not test futile cycling in any manner, have no controls or data on this point, nor further explanation, this should be removed.

3) Information in their model Figure 8) is not all that informative, nor that descriptive to the findings presented in this study or their model. This could be improved with a gene diagram (or equivalent) explaining better the different synthetic systems, reporters, and signaling circuits.

4) The authors rightly note that "insertion of new glycan strands requires endopeptidases that can cleave existing crosslinks to facilitate cell wall expansion." While the authors reference a 2008 review, a more appropriate reference might be the 2012 study from Manjula Reddy's group (Singh et al., 2012), which established endopeptidases as the long-anticipated 'space-maker' hydrolases.

5) "A RodA paralog, FtsW, provides TG activity in the context of the divisome (Gamba et al., 2009a)." The authors should also reference the 2019 study from Suzanne Walker's group (Taguchi et al., 2019), which showed transglycosylase activity for FtsW for the first time.

6) "These results suggest that the σI stress response acts in concert with WalKR to maintain activity of the elongasome by upregulating its associated autolysin, and this mechanism serves to balance the synthetic activities of the elongasome and aPBPs." The authors may want to consider being even more circumspect in their conclusion. The uninitiated reader may not appreciate the nuance of using a hyperactive variant of WalK (WalK*).

7) "Mutations that impair PG synthesis can often be rescued by growth on plates amended with 20 mM MgSO4 (Formstone and Errington, 2005)." The authors should take a moment here to explain that Mg^2+^ suppresses cell wall defects in *B. subtilis* by inhibiting autolysins (PMID 28317238), which is interesting since the authors show increased *lytE* expression in *ΔponA* cells (Figure 4E). Explaining the suppressive role of Mg^2+^ on autolysin activity would further support the authors' claim that directed LytE activity is important for cell survival (subsections “EcsAB-RasP functions through σ^I^ to sustain cell wall synthesis in the absence of aPBPs”, “σ^I^ supports elongasome function by regulating MreBH and LytE” and “Balance in the MreBH-LytE activity is essential for optimal elongasome function”).

8) Subsection “The EcsAB-RasP pathway is essential in the absence of class A PBPs”, Figure 1B: Please comment on the colony morphology of *ΔrasPΔponA* cells? The colonies look mucoidy and there appear to be suppressors.

---

## [Author Response]

Overall, the reviewers thought that the work provides important insight into how *Bacillus subtilis* is able to compensate for loss of class aPBP activity via upregulation of LytE under the control of SigI and WalRK. At the same time, we felt that several conclusions required additional experiments to be fully supported. Specifically:1) Data indicating that cefuroxime preferentially targets PBP1.

This is already established in the literature and the relevant references are cited in the subsection “Mutants defective in the EcsAB-RasP pathway are sensitive to antibiotics that inhibit aPBPs”.

2) Width measurements of single knockout controls of LytE and MreBH, and/or data on how induction (or titration) of one or the other affects width.

Once labs re-opened we returned to this question and gathered additional data for the *lytE* and *mreBH* single mutant strains. These are presented in revised Figure 5C and D.

Not essential, although both would strengthen the manuscript, are experiments assessing the potential for RasP/WalRK to control minor aPBPs and the impact of mreBH and lytE expression and their ability to bypass the aPBPs in ΔrasP/ΔsigI cells.

All of our results point to the idea that RasP works through SigI, and the relevant target is MreBH which works with LytE. The SigI and WalR regulons are known, and the targets do not include any PBPs.

We have confirmed that the CEF sensitivity in a *rasP* mutant is the same as in the strain lacking the three minor aPBPs. In addition, induction of *mreBH* rescues the *rasP* mutant, and this rescue is also seen in a strain lacking the three minor aPBPs (this is now mentioned in the paper).

Equally important, all three reviewers found the manuscript difficult to read, and felt it would be even more so for eLife readers who are not familiar with peptidoglycan biosynthesis and cell envelope stress response pathways in *B. subtilis*, or less comfortable with genetic approaches. For convenience, specific reviewer comments related to textual revisions are at the end of this letter. We would like you to pay particular attention to these comments and ensure that the revised manuscript is accessible to the broad audience that reads eLife.

We have modified the text in several places to address those sections that were deemed unclear.

1) The authors propose that activation of LytE bypasses the requirement for aPBPs through activation of the elongasome specific TGase, RodA (citing a 2016 Rudner lab paper) and presumably its associated TP enzymes. While exciting if correct, it is unclear specifically how activation of a TGase is sufficient to bypass both the TG and TP activity of the a PBPs and what this means for the growth and physiology of the cell. If there is substantial data explaining how activation of the TGase also leads to activation of its corresponding TPase, this information should be included in the text.

We apologize for the confusion. We did not mean to imply that RodA replaces class aPBPs. RodA only provides TG activity, and must work together with one of two class bPBPs. Our study indicates that this synthetic complex (RodA + bPBP) can regulated by changing the amount of MreBH^+^LytE, which promotes localized degradation of PG to allow incorporation of new strands.

2) “We next sought to identify antibiotics that, unlike MOE, inhibit aPBPs at their TP active site, and can thereby lead to futile cycling.” “This resulted in synergistic inhibition (Figure 2—figure supplement 2), consistent with the same target drug synergy model, as previously described for *E. coli* protein synthesis inhibitors (Yilancioglu, 2019) and drugs used to treat human diseases (Jia et al., 2009). Taken together, our results suggest that CEF preferentially targets PBP1”.The explanation in this section is incredibly vague and confusing, and it should be clarified. One point specifically, is they invoke the "same target drug synergy model", without definition or explanation. This should be explained better.

We appreciate that this section was not well organized and the logical flow was not clear. This entire section has been re-written for clarity. (subsection “Mutants defective in the EcsAB-RasP pathway are sensitive to antibiotics that inhibit aPBPs”).

Similarly, they introduce this section looking for "antibiotics that, unlike MOE, inhibit aPBPs at their TP active site, and can thereby lead to futile cycling Given they do not test futile cycling in any manner, have no controls or data on this point, nor further explanation, this should be removed.

This is the same section (subsection “Mutants defective in the EcsAB-RasP pathway are sensitive to antibiotics that inhibit aPBPs”) noted above and has been re-written. In fact, we did test the hypothesis of futile cycling by asking whether CEF and MOE were antagonistic. However, we found instead that their activities were synergistic, which is not consistent with futile cycling.

3) Information in their model Figure 8) is not all that informative, nor that descriptive to the findings presented in this study or their model. This could be improved with a gene diagram (or equivalent) explaining better the different synthetic systems, reporters, and signaling circuits.

We agree that the model was not as clear as it could have been. We now provide a re-envisioned Figure 8 which makes the relationships much clearer.

4) The authors rightly note that "insertion of new glycan strands requires endopeptidases that can cleave existing crosslinks to facilitate cell wall expansion." While the authors reference a 2008 review, a more appropriate reference might be the 2012 study from Manjula Reddy's group (Singh et al., 2012), which established endopeptidases as the long-anticipated 'space-maker' hydrolases.

We have carefully reviewed all citations and included the suggested citations as requested. In addition, citations have been updated were needed.

5) "A RodA paralog, FtsW, provides TG activity in the context of the divisome (Gamba et al., 2009a)." The authors should also reference the 2019 study from Suzanne Walker's group (Taguchi et al., 2019), which showed transglycosylase activity for FtsW for the first time.

We have included the suggested citation as requested.

6) "These results suggest that the σ^I^ stress response acts in concert with WalKR to maintain activity of the elongasome by upregulating its associated autolysin, and this mechanism serves to balance the synthetic activities of the elongasome and aPBPs." The authors may want to consider being even more circumspect in their conclusion. The uninitiated reader may not appreciate the nuance of using a hyperactive variant of WalK (WalK*).

This text has been revised to be more circumspect, as requested. Rather than stating that this “serves” to balance the activities, we say that it “helps to maintain balanced activity of the elongasome and the aPBPs during cell elongation.”

7) "Mutations that impair PG synthesis can often be rescued by growth on plates amended with 20 mM MgSO4 (Formstone and Errington, 2005)." The authors should take a moment here to explain that Mg^2+^ suppresses cell wall defects in *B. subtilis* by inhibiting autolysins (PMID 28317238), which is interesting since the authors show increased lytE expression in ΔponA cells (Figure 4E). Explaining the suppressive role of Mg^2+^ on autolysin activity would further support the authors' claim that directed LytE activity is important for cell survival (subsections “EcsAB-RasP functions through σ^I^ to sustain cell wall synthesis in the absence of aPBPs”, “σ^I^ supports elongasome function by regulating MreBH and LytE” and “Balance in the MreBH-LytE activity is essential for optimal elongasome function”).

We thank the reviewer for this suggestion. We now include a clearer description of how Mg supplementation works when first introduced, and specifically highlight the role in suppression of autolysin activity (subsection “The EcsAB-RasP pathway is essential in the absence of class A PBPs”).

8) Subsection “The EcsAB-RasP pathway is essential in the absence of class A PBPs”, Figure 1B: Please comment on the colony morphology of ΔrasP ΔponA cells? The colonies look mucoidy and there appear to be suppressors.

The image has been replaced. The colonies are not mucoid, but they are slow-growing and in the original image some suppressors were apparent, as the referee noted.